# Involvement of Angiogenesis in the Pathogenesis of Coronary Aneurysms

**DOI:** 10.3390/biomedicines9091269

**Published:** 2021-09-19

**Authors:** Sylwia Iwańczyk, Tomasz Lehmann, Artur Cieślewicz, Artur Radziemski, Katarzyna Malesza, Michał Wrotyński, Paweł Piotr Jagodziński, Marek Grygier, Maciej Lesiak, Aleksander Araszkiewicz

**Affiliations:** 1Department of Cardiology, Poznan University of Medical Sciences, 61-848 Poznań, Poland; michal.wrotynski@skpp.edu.pl (M.W.); marek.grygier@skpp.edu.pl (M.G.); maciej.lesiak@skpp.edu.pl (M.L.); aleksander.araszkiewicz@skpp.edu.pl (A.A.); 2Department of Biochemistry and Molecular Biology, Poznan University of Medical Sciences, 61-781 Poznań, Poland; tlehmann@ump.edu.pl (T.L.); pjagodzi@ump.edu.pl (P.P.J.); 3Department of Clinical Pharmacology, Poznan University of Medical Sciences, 61-848 Poznań, Poland; artcies@ump.edu.pl (A.C.); kmalesza@ump.edu.pl (K.M.); 4Department of Hypertensiology, Angiology and Internal Medicine, Poznan University of Medical Sciences, 61-848 Poznań, Poland; artur.radziemski@skpp.edu.pl

**Keywords:** coronary aneurysm, coronary artery disease, angiogenesis

## Abstract

The present study aimed to evaluate the plasma concentration of pro and antiangiogenic factors and their role in the pathogenesis of coronary artery abnormal dilation (CAAD). We measured the plasma concentration of matrix metalloproteinase-8 (MMP-8), transforming growth factor beta 1 (TGF-β1), Angiopoietin-2, vascular endothelial growth factor (VEGF), and fibroblast growth factor (FGF) using a sandwich ELISA technique in the plasma of patients with coronary artery abnormal dilation (CAAD, Group 1), coronary artery disease (CAD, Group 2), and normal coronary arteries (NCA, Group 3). Patients suffering from CAAD showed significantly higher plasma concentrations of VEGF (*p* = 0.002) than those from the control group. Both pathological angiogenesis and inflammation appear to be crucial in the pathogenesis of aneurysmal dilatation of the coronary arteries.

## 1. Introduction

Coronary artery abnormal dilation (CAAD) is one of the uncommon cardiovascular disorders with an incidence range of 0.15–5.3% of patients undergoing coronary angiography [1]. This pathology has been described interchangeably using two terms: coronary artery aneurysm (CAA) and coronary artery ectasia (CAE) [2]. However, these synonymously used terms refer to two different phenotypes. CAA is defined as a focal dilatation with a diameter of 1.5 times the adjacent normal coronary artery, whereas the term CAE describes similar but more diffused lesions [3]. According to the anatomical shape of the dilated segment, CAA is classified into fusiform type if the longitudinal diameter exceeds the transverse diameter or saccular type in the reverse case [4]. The most common underlying mechanism of abnormal coronary dilatation is atherosclerosis, linked to 50% of CAADs diagnosed in adults [5]. Less common causes are Kawasaki disease, diagnostic or interventional coronary angiography, inflammatory and infectious arteritis, connective tissue disease, aortic dissection, tumor metastases, trauma, and congenital malformation.

In response to advanced atherosclerosis, the local specific conditions such as anoxia, inflammation, and oxidative stress stimulate the angiogenic factors that promote sprouting angiogenesis from preexisting vasa vasorum [4]. Neovascularization promotes macrophage infiltration, lipid deposition, inflammation, and consequently atherosclerotic lesion progression [6]. In addition, apart from increasing the local flow of nutrients and oxygen, neovascularization may result in destabilization and remodeling of the atherosclerotic plaque.

Angiogenesis is regulated by a complex array of pro and antiangiogenic factors [7]. The imbalance between their effects may result in adverse remodeling and destabilization of the artery wall, including plaque instability and rupture [8,9]. The consequence of the above processes may be the weakening of the vessel wall and thus an increased risk of coronary aneurysms [10]. A previous study showed that the total aneurysm volume was positively correlated with endocan (endothelial cell-specific molecule-1—ESM-1) concentration. ESM-1 regulates both inflammatory and angiogenetic processes, being a potential marker of vascular wall damage resulting from inflammation and vascular remodeling as a result of a disturbance of pro and antiangiogenic processes [11]. Angiogenesis is also regulated by molecular mechanisms [7,12]. Recent studies have revealed that the expression profile of many miRNAs differs significantly in patients with coronary aneurysms and aortic aneurysms [13,14]. 

Our study aimed to evaluate the role of angiogenic processes in the pathogenesis of CAAD by assessing plasma levels of matrix metalloproteinase-8 (MMP-8), transforming growth factor beta 1 (TGF-β1), Angiopoietin-2, vascular endothelial growth factor (VEGF), and fibroblast growth factor (FGF) in patients with CAAD compared to those in age- and sex-matched patients with stable angina and obstructive coronary artery disease (CAD), and control subjects with angiographically normal coronary arteries (NCA).

## 2. Materials and Methods

### 2.1. Study Design and Patient Selection

Two hundred patients undergoing coronary angiography for typical chest pain or discomfort, according to the European Society of Cardiology (ESC) guidelines, were enrolled between October 2018 and March 2020 [15]. Fifty consecutive patients with coronary abnormal aneurysmal dilatation (CAAD) were included in the study Group 1. CAAD was defined as a diffused or focal dilatation of the coronary artery with a diameter of 1.5 times the adjacent normal segment. The group also included patients with CAAD and concomitant stenosis of coronary arteries. 

Fifty patients with angiographically documented CAD without CAAD were randomly selected from the overall cohort and included in Group 2. The angiographic criteria for CAD were: coronary artery stenosis >90% or intermediate stenosis (50–90%) with documented ischemia or hemodynamically significant, defined as either fractional flow reserve (FFR) ≤ 0.80 or an instantaneous wave-free ratio (iFR) ≤ 0.89. Fifty patients with the angiographic exclusion of significant coronary stenosis and dilatation (NCA) were randomly selected from the overall cohort and enrolled as control patients in Group 3. All patients in the control group presented with normal ECG and echocardiography and had no evidence of ischemia during noninvasive stress tests. Group 2 and Group 3 were matched to the study group in terms of sex and age. The exclusion criteria were as follows: (1) acute coronary syndrome; (2) elevated troponin I (TNI) or creatine kinase (CK-MB) levels; (3) history of severe hepatic and renal dysfunction; (4) leukemia, leukopenia, thrombocytopenia, or ongoing inflammatory and malignant diseases; (5) systemic diseases of connective tissue; (6) Interferon treatment; (7) no informed consent. The selection process for the study groups is demonstrated in the flowchart in Figure 1.

The institutional ethics review board approved the protocol (no. 984/18), and the study was conducted following the Declaration of Helsinki. Written informed consent was obtained from each individual.

### 2.2. ELISA Analysis

EDTA blood samples (10 mL) were collected from all patients on the first day after the cardiac catheterization procedure and were processed within 30 min of collection. Samples were centrifuged at 1300 × *g* for 15 min at room temperature. The supernatant was stored at −80 °C.

Measurements were performed in batches. MMP-8 levels were determined using an ELISA sandwich kit (Human MMP-8 ELISA Kit Catalogue Number EHMMP8, Thermo Fisher Scientific Inc., Waltham, MA, USA) according to the supplier’s instructions. Briefly, 100 µL of 10 times diluted plasma was incubated in anti-MMP-8 antibody precoated wells. After washing and aspiration, the samples were incubated with a biotin-conjugated anti-MMP-8 antibody. TGF-β1 levels were determined using an ELISA sandwich kit (Human TGF beta 1 Platinum ELISA, Catalogue Number BMS249/4, Bender MedSystems GmbH Campus Vienna Biocenter, Thermo Fisher Scientific Inc., Vienna, Austria) according to the supplier’s instructions. Briefly, 180 µL of 10 times diluted plasma was incubated for 1 h at room temperature with 20 μL of 1 N HCI. The mix was neutralized by the addition of 20 μL of 1 N NaOH and 40 µL was added into anti-TGF-β1 antibody precoated wells. After washing and aspiration, the samples were incubated with a biotin-conjugated anti-TGF-β1 antibody. 

Angiopoietin-2 levels were determined using an ELISA sandwich kit (Human Angiopoietin-2 ELISA Kit Catalogue Number KHC1641, Bender MedSystems GmbH for Thermo Fisher Scientific Inc.) according to the supplier’s instructions. Briefly, 100 µL of 10 times diluted plasma was incubated in anti-Angiopoietin-2 antibody precoated wells. After washing and aspiration, the samples were incubated with a biotin-conjugated anti-Angiopoietin-2 antibody. VEGF levels were determined using an ELISA sandwich kit (Human VEGF ELISA Kit, Catalog Number KHG0111, Bender MedSystems GmbH for Thermo Fisher Scientific Inc.) according to the supplier’s instructions. Briefly, 100 µL of 20 times diluted plasma was incubated in anti-VEGF antibody precoated wells. After washing and aspiration, the samples were incubated with a biotin-conjugated anti-VEGF antibody. FGF levels were determined using an ELISA sandwich kit (Human FGF basic ELISA Kit, Catalog Number KHG0021, Bender MedSystems GmbH for Thermo Fisher Scientific Inc.), according to the supplier’s instructions. Briefly, 100 µL of 20 times diluted plasma was incubated in anti-FGF antibody precoated wells. After washing and aspiration, the samples were incubated with a biotin-conjugated anti-FGF antibody. Each type of analyzed angiogenic factor, MMP-8, TGF-β1, VEGF, Angiopoietin-2, and FGF was processed subsequently according to the same procedure. The streptavidin-HRP solution was added to each well. The quantity of peroxidase bound to each well was determined by adding tetramethylbenzidine (TMB) substrate. The reaction was stopped, and the resultant color was read at 450 nm in an Epoch Microplate Spectrophotometer (Biotek, Winooski, VT). The concentration of analyzed protein in the sample was determined by interpolation from the standard curve.

### 2.3. Statistical Analysis

Standard descriptive statistics were used in the analysis. Continuous variables are presented as an arithmetic mean with standard deviation (for normal distribution) or median with interquartile range (IQR) (for non-normal distribution). The normality of the distribution of variables was tested using the Kolmogorov–Smirnov test. Categorized variables are presented in absolute numbers and percentages. The significance of differences between the mean values of the continuous data was assessed using a one-way analysis of variance (ANOVA) for multiple comparison. The post hoc Tukey’s honestly significant difference test (Tukey’s HSD) was performed to make all of the pairwise comparisons between groups. The Kruskal–Wallis test was used to compare continuous variables with a distribution deviating from the normal. The frequency of occurrence of categorized variables was calculated using the chi-squared test. PQStat Software (Poznan, Poland) was used for statistical analysis.

## 3. Results

### 3.1. Clinical Characteristics of the Study Population

The clinical characteristics of the study population are summarized in Table 1. Patients with CAAD had a significantly higher BMI than patients with CAD (*p* = 0.005). The control patients had no history of cardiovascular events and were characterized by a significantly lower incidence of hypertension than others (*p* = 0.04). Moreover, a significantly lower percentage of them were treated with aspirin (*p* < 0.001) and beta blockers (*p* = 0.009). No patient in the control group was indicated for dual antiplatelet therapy (DAPT) (*p* < 0.001). Statins were used less in the control group than in patients with atherosclerosis (*p* = 0.008). Clopidogrel was most commonly used in patients with CAD (*p* = 0.03). C-reactive protein (CRP) was comparable in all groups (*p* = 0.3).

### 3.2. Angiographic Characteristics of Group 1 (CAAD Group)

The angiographic characteristics of Group 1 are presented in Table 2. CAE was diagnosed in the majority of patients (75%). Abnormal dilatation of the coronary artery, both diffused and focal, was most often localized in the right coronary artery (RCA) and involved only one vessel. CAD was angiographically documented (significant stenosis or previous coronary intervention) in 55% of patients.

### 3.3. ELISA Analysis of Angiogenesis Factors

An ANOVA test demonstrated that the VEGF level differed significantly between the study groups (*p* = 0.04) (Figure 2). Post hoc Tukey’s HSD analysis showed a significantly higher level of VEGF in the CAAD group than in the controls (*p* < 0.002). The nonparametric Kruskal–Wallis test for multiple comparisons was used to analyze the other biomarkers. It revealed that TGF-β1 levels differ significantly between the groups (*p* = 0.04). The post hoc analysis showed a significantly higher level of TGF-β1 in the CAD group than in the control group (*p* = 0.03). Moreover, a trend towards higher TGF-β1 levels in the CAAD vs. control patients was observed (*p* = 0.06). The levels of other angiogenic factors, i.e., MMP-8, Angiopoietin-2, and FGF, were comparable in all studied groups (Table 3, Figure 2). However, we showed a trend towards a higher level of MMP-8 in the CAAD group compared to the CAD group (*p* = 0.06).

## 4. Discussion

The present study demonstrates that the VEGF level was significantly higher in the CAAD group than in the control group. We also identified a trend towards higher MMP-8, FGF, and lower TGF-β1 levels in patients with CAAD compared to the other groups. 

VEGF is an endothelial cell-specific cytokine and heparin-binding glycoprotein, recognized as a key factor in normal and pathological angiogenesis [16]. It induces the development and progression of certain pathological conditions, such as tumor growth and metastasis, macular degeneration, diabetes retinopathy, ischemic processes (myocardial ischemia), pre-eclampsia, etc. [17,18,19]. In pathological angiogenesis, VEGF mobilizes inflammatory cells to the site of injury, sustaining the local inflammatory process and stimulating the synthesis of proangiogenic factors by endothelial cells, platelets, smooth muscle cells, inflammatory cells, fibroblasts, and tumor cells [20,21,22,23,24,25,26,27]. The major trigger inducing angiogenesis is hypoxia, but other factors may also be responsible: hypoglycemia, hypertension, low pH, mechanical stress, chronic inflammation, etc. [28]. Hypoxic tissues release the hypoxia-inducible factor-1 (HIF-1), which activates the transcription of proangiogenic factors, including VEGF, basic fibroblast growth factor (bFGF, FGF-2) [29,30,31], angiopoietin-1 (Ang- 1), angiopoietin-2 (Ang-2) [28], and TGF-β [20].

Increased VEGF secretion is induced by proangiogenic factors released from injured tissues through paracrine and autocrine mechanisms [31,32]. Therefore, the normal value of CRP in our study does not exclude local inflammation within the arterial wall, e.g., as a result of an atherosclerotic process, as a factor stimulating the increase in VEGF levels.

VEGF as an angiogenic factor is responsible for the degeneration of the arterial wall and induces the synthesis of matrix metalloproteinases (MMPs) [33]. The secretion and activation of MMPs and inhibition of tissue inhibitors of metalloproteinases-2 (TIMP-2) induced by Ang-1 facilitate the decomposition of ECM, which most likely contributes to the progression of the coronary aneurysm [34,35]. MMPs regulate the remodeling of the extracellular matrix due to, among other things, the degradation of connective tissue. Research to date has revealed that MMPs play an essential role in angiogenesis, stimulating cell migration through the proteolysis of matrix components [36]. Recent evidence suggests a pathophysiological role for VEGF and MMP in positive external vascular remodeling and aneurysm formation in various vascular beds. Elevated plasma concentrations of MMP-3 and MMP-9 have been demonstrated in patients with abdominal aortic aneurysms [37,38,39].

Moreover, several authors have identified increased levels of VEGF, MMP-2, and MMP-9 in the blood of patients with abdominal aortic aneurysms [40,41,42]. Interestingly, elevated plasma levels of MMP-9, TIMP-1, and VEGF have also been found in the acute phase of Kawasaki disease, but not later, even in the presence of persistent coronary aneurysms [43,44,45]. In our study, we observed a trend towards higher MMP-8 values in the CAAD group. However, probably due to the small group of patients, statistical significance was not demonstrated.

Savino et al. showed higher VEGF levels in patients with diffuse coronary ectasia (DCE) than in stable angina (SA) or normal coronary arteries (NCA), and lower tissue inhibitors of MMP-2 (TIMP-2) levels in DCE and SA than in NCA [46]. Moreover, the authors revealed that symptomatic patients with DCE typically present with an acute coronary syndrome and exhibit a lack of obstructive stenosis at angiography, decreased plasma levels of TIMP-2, and raised plasma levels of VEGF. They concluded that the simultaneous occurrence of reduced MMPs inhibition and increased angiogenetic activity suggests an accelerated and persistent extracellular matrix remodeling process favoring arterial remodeling and aneurysms formation.

Our results showed only a trend towards higher FGF and lower TGF-β1 levels in patients with CAAD than in the other groups. We know from the literature that bFGF or FGF-2 was the first described proangiogenic factor, playing an important role in pathological angiogenesis through, among other things, mitogenic effect on endothelial cells, increase in vascular permeability, participation in tubulogenesis, or proteolytic degradation of ECM. It also stimulates the proliferation of fibroblasts, promoting granulation tissue formation and wound healing [20,47]. 

## 5. Conclusions

The results suggest that pathological angiogenesis and inflammation appear to be crucial in the pathogenesis of aneurysmal dilatation of the coronary arteries. The elevated level of plasma VEGF was found in patients with coronary artery abnormal dilation, and appears to play an important role in the pathogenesis of CAAD.

## Figures and Tables

**Figure 1 biomedicines-09-01269-f001:**
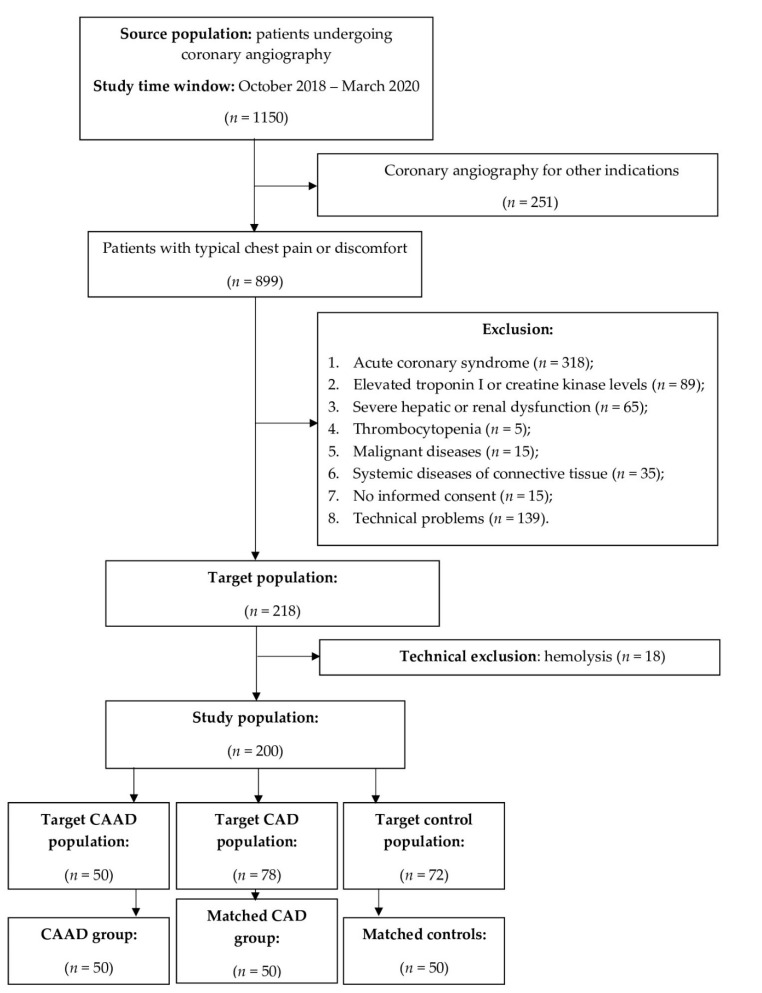
Flowchart of the study population. Inclusion and exclusion criteria for the study cohort and selection process for the coronary artery abnormal dilation (CAAD) group, coronary artery disease (CAD) group, and normal coronary arteries (NCA, control) group.

**Figure 2 biomedicines-09-01269-f002:**
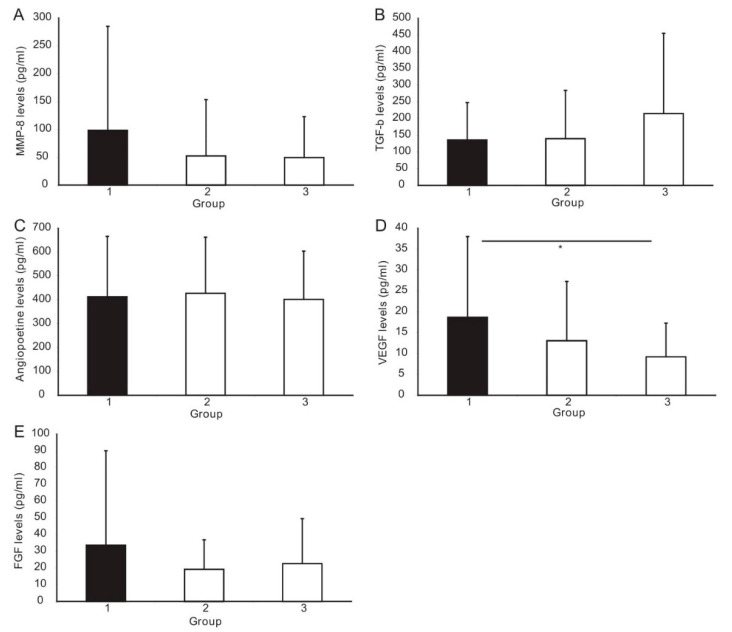
Plasma levels of the investigated angiogenic factors: (**A**) Matrix metalloproteinase-8 (MMP-8); (**B**) Transforming growth factor beta 1 (TGF-β1); (**C**) Angiopoetine-2; (**D**) Vascular endothelial growth factor (VEGF); (**E**) Fibroblast growth factors (FGF) in patients with coronary artery abnormal dilation (CAAD, Group 1) compared to patients with obstructive coronary artery disease (Group 2), and controls with normal coronary arteries (Group 3). * indicates a significant difference between the groups (*p* ≤ 0.05).

**Table 1 biomedicines-09-01269-t001:** Baseline clinical characteristic.

Baseline Data	Group 1 (*n* = 50)	Group 2 (*n* = 50)	Group 3 (*n* = 50)
Sex, male, n (%)	40 (80.0)	40 (80.0)	40 (80.0)
Age (yrs), mean ± SD	65.8 ± 8.6	66.4 ± 7.9	63.5 ± 10.1
BMI (kg/m^2^), mean ± SD	31.4 + 4.7 *	28.6 ± 4.8 *	29.1 ± 6.1
Previous MI, n (%)	16 (32.0) **^,^***	19 (38.0) **^,^***	0 **^,^***
Previous PCI, n (%)	19 (38.0) **^,^***	27 (54.0) **^,^***	0 **^,^***
Previous CABG, n (%)	3 (6.0) **^,^***	6 (12.0) **^,^***	0 **^,^***
Hypertension, n (%)	45 (90.0) **^,^***	46 (92.0) **^,^***	39 (78.0) **^,^***
Heart failure, n (%)	19 (38.0)	19 (38.0)	12 (24.0)
LVEF (%), mean ± s.d.	53.7 ± 11.9	52.8 ± 10.6	55.6 ± 8.6
Diabetes mellitus, n (%)	17 (34.0)	14 (28.0)	13 (26.0)
Hyperlipidemia, n (%)	38 (76.0)	40 (80.0)	34 (68.0)
Cigarette smoking, n (%)	20 (40.0)	18 (36.0)	15 (30.0)
Aortic aneurysm, n (%)	9 (18.0)	9 (18.0)	8 (16.0)
CKD class ≥ 2, n (%)	9 (18.0)	7 (14.0)	4 (8.0)
**Drug administration**			
Statin, n (%)	46 (92.0)	49 (98.0) ***	41 (82.0) ***
CCB, n (%)	22 (44.0)	20 (40.0)	20 (40.0)
Beta blocker, n (%)	44 (88.0) **^,^***	46 (92.0) **^,^***	36 (72.0) **^,^***
Aspirin, n (%)	43 (86.0) **^,^***	46 (92.0) **^,^***	27 (54.0) **^,^***
Clopidogrel, n (%)	23 (46.0) **^,^***	35 (70.0) **^,^***	0 **^,^***
ACEI/ARB, n (%)	44 (88.0)	45 (90.0)	39 (78.0)
**Laboratory tests**			
LDL cholesterol (mmol/L)	2.3 ± 1.7	2.2 ± 0.8	2.4 ± 0.9
CRP (mg/L)	1.7 (1.25–2.35)	2.9 (2.8–3.0)	2.1 (1.8–2.9)

ACEI—angiotensin-converting enzyme inhibitor; ARB—angiotensin II receptor blocker; BMI—body mass index; CABG—coronary artery bypass graft; CCB—calcium channel blocker; CKD—chronic kidney disease; CRP—C-reactive protein; HGB—hemoglobin; LDL—low-density lipoprotein; LVEF—left ventricle ejection fraction; MI—myocardial infarction; PCI—percutaneous coronary intervention; SD – standard deviation; WBC—white blood cell; * significant difference between Group 1 and Group 2; ** significant difference between Group 1 and Group 3; *** significant difference between Group 2 and Group 3; significant difference = *p* < 0.05.

**Table 2 biomedicines-09-01269-t002:** Angiographic characteristics of Group 1 (CAAD group).

Baseline Data	Group 1 (*n* = 50), *n* (%)
CAE	37 (74.0)
CAA	10 (20.0)
Both	3 (6.0)
Number of vessels involved	
1	35 (70.0)
2	12 (24.0)
3	3 (6.0)
Vessel localization	
LM	3 (6.0)
RCA	25 (50.0)
LAD	20 (40.0)
LCx	20 (40.0)
Concomitant CAD	28 (56.0)

CAA—coronary artery ectasia; CAD—coronary artery disease; CAE—coronary artery aneurysm; LAD—left artery descending; LCx—left circumflex artery; LM—left main; RCA—right coronary artery.

**Table 3 biomedicines-09-01269-t003:** Plasma levels of angiogenic factors in patients with coronary artery abnormal dilation (CAAD, Group 1) in comparison to patients with obstructive coronary artery disease (Group 2), and controls with normal coronary arteries (Group 3).

Angiogenic Factors	Group 1	Group 2	Group 3	1 vs. 2	1 vs. 3	2 vs. 3
MMP-8, pg/mL	40.7 (23.1–87.4)	15.6 (12.4–42.1)	31.7 (11.3–56.8)	0.06	0.58	0.58
TGF-β1, pg/mL	114.2 (42.2–184.0)	77.1 (40.3–217.0)	139.0 (82.9–253.6)	0.48	0.06	0.03
Angiopoietin-2, pg/mL	325.4 (252.2 457.0)	336.2 (283.8–492.3)	361.8 (273.8–486.5)	0.94	0.94	0.94
VEGF, pg/mL	18.7 ± 19.3	13.1 ± 14.2	9.2 ± 8.1	0.21	0.002	0.51
FGF, pg/mL	14.9 (8–30)	14.9 (7.5–29)	12.7 (4–33.5)	0.97	0.99	0.98

FGF—Fibroblast growth factors; MMP-8—matrix metalloproteinase-8; TGF-β1—transforming growth factor beta 1; VEGF—vascular endothelial growth factor; *p*-values ≤ 0.05 were considered significant.

## Data Availability

Data available on request due to privacy and ethical restrictions.

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
