# Peer review of "Involvement of Angiogenesis in the Pathogenesis of Coronary Aneurysms"

_biomedicines, 2021, doi:10.3390/biomedicines9091269_

Round 1
Reviewer 1 Report
Iwańczyk et al. conduct a cross-sectional study to compare plasma concentrations of 5 angiogenic factors between patients with coronary artery abnormal dilation, patients with stable angina and obstructive coronary artery disease, and normal controls. They demonstrated a higher level of VEGF among patients with coronary artery abnormal dilation than patients with stable angina and obstructive coronary artery disease or normal controls. Several study concerns should be addressed.
1. Please provide the document no. of ethical permission in the method section.
2. A flowchart should be provided to demonstrate the selection process for the CAAD group (group 1), CAD without CAAD group (group 2), and controls patients (group 3). Since these patients were selected from the overall cohort, clear inclusion and exclusion criteria should be presented in this study using a flowchart.
3. The blood samples were collected on the first day after the cardiac catheterization procedure. Thus, all angiogenic factors were the blood level after cardiac catheterization. Some studies suggested periprocedural biomarkers may exist [1, 2]. Is there any biomarker data before the cardiac catheterization procedure?
Reference
[1] JACC Cardiovasc Interv. 2019 Oct 14;12(19):1954-1962
[2] Am J Cardiol. 2006 Oct 1;98(7):915-7
4. Several angiogenic factors could be found in the literature. How to select these 5 biomarkers in this study?
5. Please check the study rationale. The researcher includes 3 groups in this study. In general, the demonstration of dose-dependent effect between groups or cases compared to control 1 or control 2 will be proposed to show the results. In Table 2, how to compare the angiogenic factors between groups? For the symbol “ * ”, which groups were compared here? It seems a higher level of VEGF in Group 1 than Group 2 and a higher level of VEGF in Group 2 than Group 3. However, the researcher only compared the plasma concentrations of VEGF between Group 1 and Group 3 in multivariable analysis with confounders adjustment. If a comparison between 3 groups (Group 1 vs. Group 2 vs. Group 3), multinomial logistic regression or ordinal logistic regression could be considered.
6. In the multivariable analysis, only cardiovascular risk factors (diabetes mellitus, hypertension, and dyslipidemia) were adjusted in the model. Medications may also influence the angiogenic factors (exposure) and group differences (outcome; CAAD group and control group). Besides, several medications differences could be observed in Table 1, but no further control for these confounders. How to convince the reader to evaluate the association between angiogenic factors and the CAAD group without control medications?
Reviewer 2 Report
Journal: Biomedicines (ISSN 2227-9059)
Manuscript ID: biomedicines-1332697
Title: Involvement of angiogenesis in the pathogenesis of coronary aneurysms.
The present study aimed to evaluate the plasma concentration of pro and anti-angiogenic factors and their role in the pathogenesis of coronary artery abnormal dilation (CAAD). We measured the plasma concentration of matrix metalloproteinase-8 (MMP-8), transforming growth fac-tor-beta 1 (TGF-β1), Angiopoietin-2, vascular endothelial growth factor (VEGF), and fibroblast growth factor (FGF) using a sandwich ELISA technique in plasma of patients with coronary ar-tery abnormal dilation (CAAD, Group 1), coronary artery disease (CAD, Group 2) and normal coronary arteries (NCA, Group 3). Patients suffering from CAAD showed significantly higher plasma concentrations of VEGF (p = 0.04) than CAD and control groups. The level of VEGF was not related to cardiovascular risk factors like diabetes, dyslipidemia, and hypertension (p = 0.04). Elevated level of plasma VEGF was found in patients with coronary artery abnormal dilation, and it seems to be a prognostic factor in CAAD compared to control patients. This effect is independent of other concomitant risk factors. It appears that both, pathological angiogenesis and inflammation, appear to be crucial in the pathogenesis of aneurysmal dilatation of the coronary arteries.
This is an interesting manuscript. I encourage the authors to consider the following points.
1) Methodology. Although I am not an expert in statistics, I would like the authors to review whether the test employed are correct. In the methods, the authors perform student´s T test or Man Whitney test and I believe those test are adequate for comparison of continuous variables in two groups, however, comparisons in the manuscript are made in three groups. I think ANOVA and KRUSKALL-WALLIS tests should be performed.
2) Please reference CAAD definition. Which is the difference with coronary ectasia?. I would like the authors to elaborate on this point.
3) Multivariate analysis. I suggest the authors to remove this information, as in my opinion is rather confusing and adds nothing to the understanding of the relation of biomarkers and CAAD.
Round 2
Reviewer 1 Report
I agree to compare different groups by ANOVA test and post-hoc analysis. However, the results could not demonstrate the independent association between angiogenesis biomarkers and coronary artery phenotype without confounders adjustment. Therefore, the study limitation should be acknowledged in the discussion section.